# Interpretable Node Representation with Attribute Decoding

**Xiaohui Chen**[*]                                                      *xiaohui.chen@tufts.edu*
*Department of Computer Science*
*Tufts University*

**Xi Chen**[*]                                                             *xi.chen15@rutgers.edu*
*Department of Computer Science*
*Rutgers University*

**Li-Ping Liu**                                                            *liping.liu@tufts.edu*
*Department of Computer Science*
*Tufts University*

**Reviewed on OpenReview:** *https: // openreview. net/ forum? id= AZIfC91hjM*

## Abstract

Variational Graph Autoencoders (VGAEs) are powerful models for unsupervised learning of node representations from graph data. In this work, we systematically analyze modeling node attributes in VGAEs and show that attribute decoding is important for node representation learning. We further propose a new learning model, interpretable NOde Representation with Attribute Decoding (NORAD). The model encodes node representations in an interpretable approach: node representations capture community structures in the graph and the relationship between communities and node attributes. We further propose a rectifying procedure to refine node representations of isolated notes, improving the quality of these nodes' representations. Our empirical results demonstrate the advantage of the proposed model when learning graph data in an interpretable approach.

## 1 Introduction

Graph data are ubiquitous in real-world applications. Graph data contain rich information about graph nodes. The community structure among graph nodes is particularly interesting. Such structures are modeled by traditional models such as the stochastic blockmodel (SBM) (Wang & Wong, 1987; Snijders & Nowicki, 1997) and its variants, which assign a node to one or multiple communities.

Learning node representations are widely investigated in document networks. One typical branch is Relational Topic Model (RTM) (Chang & Blei, 2009) and its variants (Panwar et al., 2021; Bai et al., 2018). In such models, the links and node attributes are decoded from the node representations, which are learned from documents with rich textual information. RTM-based models often provide interpretable representations due to their carefully-designed graphical models. However, the effectiveness of the RTM-based models highly relies on the text richness of the dataset.

On par with the RTM-based models, Variational Graph Autoencoder (VGAE) (Kipf & Welling, 2016) utilizes Graph Neural Networks (GNNs) to learn node vectors that encode information that can be used for reconstructing the graph structure. This model is further improved in multiple aspects. Mehta et al. (2019); Li et al. (2020); Sarkar et al. (2020) adopt different priors to learn node representations for different applications. Hasanzadeh et al. (2019) introduce a hierarchical variational framework to VGAE and modify the link decoder. Pan et al. (2018; 2019); Tian et al. (2022b) adopt the training scheme of generative adversarial networks (GANs) (Goodfellow et al., 2014) to encode the node latent representations. Tian et al. (2022a) employs

---

[*]Equal contribution.

a masked autoencoder to learn node representations for heterogeneous graphs. Following VGAE, most of its variants only use node attributes in the encoder but do not decode node attributes. Though some other models (Mehta et al., 2019; Cheng et al., 2021) use separate decoder heads to generate graph edges and node attributes. There is no systematic approach to analyzing how node attributes should be used to learn representations in the VGAE framework better.

In this work, we analyze VGAE using the information-theoretic framework by Alemi et al. (2018) and show the theoretical strengths of different model constructions. The analysis indicates that appropriate modeling of node attributes benefits a large class of link prediction tasks. Node attributes not only help to overcome the limitation of structure node representations by breaking symmetric structures in the graph (Srinivasan & Ribeiro, 2019), but also provide rich information for link prediction.

We further devise a new representation learning model, interpretable Node Representation with Attribute Decoding (NORAD), which learns interpretable node representations from the graph data. To exploit node attributes, the model includes a specially designed decoder for node attributes. The model can learn good representations for nodes with low degrees. When there are isolated nodes in the graph, their representations are generally hard to learn. We show that their representations can be better learned by attribute decoding. We also propose a rectification procedure to refine representations of isolated nodes.

We conduct extensive experiments to evaluate and diagnose our model. The results show that node representations learned by our model perform well in link prediction and node clustering tasks, indicating the good quality of these representations. We also show that the learned node representations capture community structures in the graph and the relationship between communities and node attributes.

Our contributions can be summarized as follows:

- we systematically examine VGAE through an information-theoretic analysis;
- we propose a new model NORAD, which includes a specially designed attribute decoder and a refinement procedure for representations of isolated nodes; and
- we conduct extensive experiments to study the quality and interpretability of node representations learned by NORAD.

## 2 Preliminaries

Let $G = (\mathbf{A}, \mathbf{X})$ denote an attributed graph with $n$ nodes, $\mathbf{A} \in \mathbb{R}^{n \times n}$ is the binary adjacency matrix of the graph, and $\mathbf{X} = (\mathbf{x}_i)_{i=1}^n \in \mathbb{R}^{n \times D}$ denotes node attributes, with $\mathbf{x}_i$ being the attribute vector of node $i$. We consider the problem of jointly modeling $\mathbf{A}$ and $\mathbf{X}$. The goal is to learn interpretable node representations $\mathbf{Z} = (\mathbf{z}_i)_{i=1}^n \in \mathbb{R}^{n \times K}$ that can better explain the data. Then $\mathbf{Z}$ provides essential information for downstream tasks such as node clustering.

**Graph Neural Networks.** GNN is a type of neural network designed to extract information from graph data. It typically consists of multiple layers, each of which runs a message-passing procedure to encode information into a node's vector representation. Let $\mathbf{H} = \text{gnn}(\mathbf{A}, \mathbf{X}; \phi)$ denote the network function of an $L$-layer GNN, which is typically defined by $\mathbf{H}^{(0)} = \mathbf{X}$,

$$\mathbf{H}^{(l)} = \sigma\left(\mathbf{H}^{(l-1)}\mathbf{W}^{(l)} + \mathbf{A}\mathbf{H}^{(l-1)}\mathbf{V}^{(l)}\right), \quad l = 1, \ldots, L,$$

and $\mathbf{H} = \mathbf{H}^{(L)}$. Here $\sigma(\cdot)$ is the activation function. $\mathbf{W}^{(l)}$ and $\mathbf{V}^{(l)}$ are the network weights for the $l$-th layer. We denote all network weights with $\phi$.

**Variational Graph Autoencoder.** VGAE (Kipf & Welling, 2016) learns node representations in an unsupervised approach based on variational auto-encoder (VAE) (Kingma & Welling, 2013). In VGAE, the prior distribution $p(\mathbf{Z})$ over node presentations $\mathbf{Z}$ is a standard Gaussian distribution. And the generative

model $p(\mathbf{A}|\mathbf{Z})$ is defined as

$$p(\mathbf{A}|\mathbf{Z}) = \prod_{i=1}^{n} \prod_{j=1}^{n} p(A_{ij}|\mathbf{z}_i, \mathbf{z}_j), \quad p(A_{ij} = 1|\mathbf{z}_i, \mathbf{z}_j) = \text{sigmoid}(\mathbf{z}_i^\top \mathbf{z}_j). \tag{1}$$

VGAE is a type of variational autoencoder (Kingma & Welling, 2013). It defines its variational distribution $q(\mathbf{Z}|\mathbf{A}, \mathbf{X})$ to be a Gaussian distribution Gaussian$(\mathbf{Z}; \boldsymbol{\mu}, \boldsymbol{\sigma}^2)$, which is defined by a GNN.

$$(\boldsymbol{\mu}, \boldsymbol{\sigma}) = \text{gnn}(\mathbf{A}, \mathbf{X}). \tag{2}$$

The output of the GNN is split into $\boldsymbol{\mu}$ and $\boldsymbol{\sigma}$, representing the mean and standard derivation, respectively. VGAE maximizes the Evidence Lower Bound (ELBO) of the marginal log-likelihood $\log p(\mathbf{A})$ to learn the encoder and the decoder:

$$\mathcal{L}_g = \mathbb{E}_{q(\mathbf{Z}|\mathbf{A}, \mathbf{X})} \big[ \log p(\mathbf{A}|\mathbf{Z}) + \log p(\mathbf{Z}) - \log q(\mathbf{Z}|\mathbf{A}, \mathbf{X}) \big]. \tag{3}$$

After the encoder is learned, its mean $\boldsymbol{\mu}$ can be used as deterministic node representations. Note that node features $\mathbf{X}$ are not decoded in this model. Therefore, $\boldsymbol{\mu}$ mostly represents structural information of nodes but not attribute information, though the $\boldsymbol{\mu}$ is also computed from $\mathbf{X}$.

## 3   An Information-Theoretic Analysis of Attribute Decoding

In this section, we analyze VGAE from the perspective with the rate-distortion theory (Alemi et al., 2018) and consider the mutual information between the encoding $\mathbf{Z}$ and observed data $(\mathbf{A}, \mathbf{X})$. Let $p_*(\mathbf{A}, \mathbf{X})$ be the data distribution and $H$ be its entropy, which is a constant. Let $I_q = I[(\mathbf{A}, \mathbf{X}); \mathbf{Z}]$ denote the mutual information between $(\mathbf{A}, \mathbf{X})$ and $\mathbf{Z}$. Note that $I_q$ is defined from the encoder distribution $q(\mathbf{Z}|\mathbf{A}, \mathbf{X})$ and the data distribution. The maximization of the ELBO

$$\mathcal{L}_e = \mathbb{E}_{q(\mathbf{Z}|\mathbf{A}, \mathbf{X})} \big[ \log p(\mathbf{A}, \mathbf{X}|\mathbf{Z}) + \log p(\mathbf{Z}) - \log q(\mathbf{Z}|\mathbf{A}, \mathbf{X}) \big] \tag{4}$$

can be viewed as indirect maximization of the mutual information $I[(\mathbf{A}, \mathbf{X}); \mathbf{Z}]$ under a rate constraint (Alemi et al., 2018). We further decompose $I[(\mathbf{A}, \mathbf{X}); \mathbf{Z}]$ as follows:

$$I[(\mathbf{A}, \mathbf{X}); \mathbf{Z}] = I[\mathbf{A}; \mathbf{Z}] + I[\mathbf{X}; \mathbf{Z}] + I[\mathbf{A}; \mathbf{X}|\mathbf{Z}] - I[\mathbf{A}; \mathbf{X}] \tag{5}$$

In this decomposition, the last term $I[\mathbf{A}; \mathbf{X}]$ is a constant decided by the data. The first term is the information about $\mathbf{A}$ from $\mathbf{Z}$, and the second is the information about $\mathbf{X}$. The third term $I[\mathbf{A}; \mathbf{X}|\mathbf{Z}]$ is the information between $\mathbf{A}$ and $\mathbf{X}$ that cannot be explained by $\mathbf{Z}$. When $\mathbf{Z}$ is lossless encoding, $I[\mathbf{A}; \mathbf{X}|\mathbf{Z}] = 0$.

The decomposition above is derived from the encoder distribution $q(\mathbf{Z}|\mathbf{A}, \mathbf{X})$. Still, it helps us to design the decoder $p(\mathbf{A}, \mathbf{X}|\mathbf{Z})$, which approximates $q(\mathbf{A}, \mathbf{X}|\mathbf{Z})$ when maximizing the ELBO $\mathcal{L}_e$. One variant of VGAE (Mehta et al., 2019; Cheng et al., 2021) assumes $p(\mathbf{A}, \mathbf{X}|\mathbf{Z}) = p(\mathbf{X}|\mathbf{Z})p(\mathbf{A}|\mathbf{Z})$. In this case, the conditional mutual information $I[\mathbf{A}; \mathbf{X}|\mathbf{Z}]$ tends to be zero. The conditional independence assumption may require $\mathbf{Z}$ to have extra bits so that $\mathbf{Z}$ can explain $\mathbf{A}$ and $\mathbf{X}$ separately. In this model choice, the lower bound becomes

$$\mathcal{L}_a = \mathbb{E}_{q(\mathbf{Z}|\mathbf{A}, \mathbf{X})} \big[ \log p(\mathbf{A}|\mathbf{Z}) + \log p(\mathbf{X}|\mathbf{Z}) + \log p(\mathbf{Z}) - \log q(\mathbf{Z}|\mathbf{A}, \mathbf{X}) \big]. \tag{6}$$

Other possible variants that do not have the assumption, the decoder $p(\mathbf{A}, \mathbf{X}|\mathbf{Z})$ can be decomposed as $p(\mathbf{X}|\mathbf{Z})p(\mathbf{A}|\mathbf{X}, \mathbf{Z})$, then it is more flexible and should fit the data better. However, $\mathbf{Z}$ is less capable of explaining $\mathbf{A}$ because part of the information is encoded in $\mathbf{X}$. A similar issue happens in the decomposition $p(\mathbf{A}, \mathbf{X}|\mathbf{Z}) = p(\mathbf{A}|\mathbf{Z})p(\mathbf{X}|\mathbf{A}, \mathbf{Z})$.

Another variant, Graphite (Grover et al., 2019), takes a totally different approach: it only lets $\mathbf{Z}$ encode $\mathbf{A}$ and conditions the entire model on $\mathbf{X}$. Therefore, the ELBO is

$$\mathcal{L}_c = \mathbb{E}_{q(\mathbf{Z}|\mathbf{A}, \mathbf{X})} \big[ \log p(\mathbf{A}|\mathbf{X}, \mathbf{Z}) + \log p(\mathbf{Z}|\mathbf{X}) - \log q(\mathbf{Z}|\mathbf{A}, \mathbf{X}) \big] \leq I[\mathbf{A}; \mathbf{Z}|\mathbf{X}]. \tag{7}$$

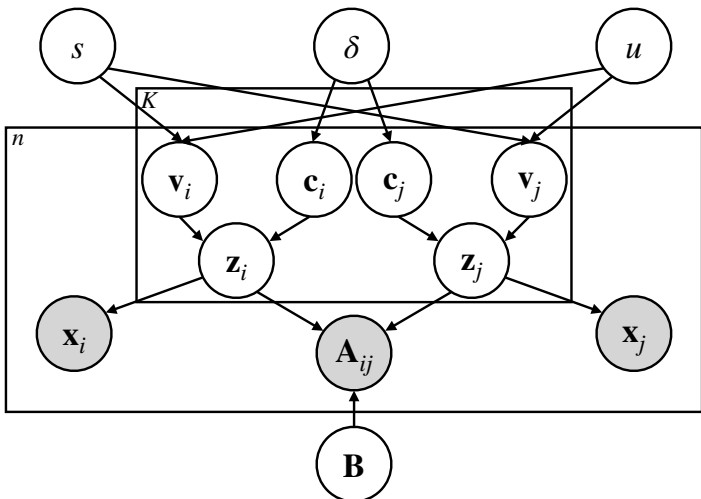

Figure 1: The plate representation of our framework. Here $(\delta, u, s)$ are the prior parameters.

In this case, the optimal encoding $\mathbf{Z}$ only encodes the part of $\mathbf{A}$ that cannot be explained by $\mathbf{X}$. This results in $\mathbf{Z}$ only containing a small portion of the information about $\mathbf{A}$.

In the ELBO $\mathcal{L}_g$ of the basic VGAE, there is no decoding of $\mathbf{X}$, which means that the encoder has no incentive to encode any information about $\mathbf{X}$.

Our analysis shows that the simple formulation $\mathcal{L}_a$ with conditional independence assumption is better if we want $\mathbf{Z}$ to maximally encode information about $(\mathbf{A}, \mathbf{X})$. Though we are not the first to discover this formulation, our analysis of different formulations helps deepen the understanding of VGAE variants.

**Fix the bias of model fitting in link prediction.** Node representations of VGAE are often used in link prediction tasks. Therefore, the given adjacency matrix is $\hat{\mathbf{A}}$, which is sparser than the true adjacency matrix $\mathbf{A}$. In this case, we must do model fitting with $\hat{\mathbf{A}}$ instead of $\mathbf{A}$.

Model training based on $\hat{\mathbf{A}}$ is biased because actual edges missing from $\hat{\mathbf{A}}$ are mixed with non-edges. This issue has been studied by Liang et al. (2016); Liu & Blei (2017) in different contexts. Specifically, the bias is due to the term $E_q\big[\log p(\hat{\mathbf{A}}|\mathbf{Z})\big]$, which leads $\mathbf{Z}$ to treat missing edges and non-edges equally. However, in some datasets, $\mathbf{X}$ and $\mathbf{A}$ are strongly correlated, and $\mathbf{X}$ is observed for all graph nodes. In this case, encoding information in $\mathbf{X}$ helps improve the link prediction performance, so we consider a modified ELBO:

$$\mathcal{L}_\alpha = \mathbb{E}_{q(\mathbf{Z}|\mathbf{A},\mathbf{X})}\big[\log p(\hat{\mathbf{A}}|\mathbf{Z}) + \alpha \log p(\mathbf{X}|\mathbf{Z}) + \log p(\mathbf{Z}) - \log q(\mathbf{Z}|\mathbf{A},\mathbf{X})\big]. \tag{8}$$

In a link prediction task, we set the ratio factor $\alpha > 1$ so that model training can pay more attention to node features. Note that this modified ELBO is still a lower bound of $I_q - H$ when $\mathbf{X}$ is discrete and $\log p(\mathbf{X}|\mathbf{Z}) < 0$. If we set $\alpha = 0$, it is equivalent to VGAE. If $\hat{\mathbf{A}} = \mathbf{A}$ in a representation learning task, we still use the normal ELBO and set $\alpha = 1$.

## 4 Interpretable Node Representation with Attribute Decoding

In this section, we introduce our new model, NORAD, to learn node representations. The proposed model has two goals. The primary goal is to learn high-quality node representations that best preserve the information of the attributed graph, and the secondary goal is to make the learned representations interpretable.

We will use the formulation $\mathcal{L}_\alpha$ in Equation (8) to construct our model. We need to specify four distributions: the encoder $q(\mathbf{Z}|\mathbf{A}, \mathbf{X})$, the prior $p(\mathbf{Z})$, the decoder $p(\mathbf{A}|\mathbf{Z})$, and the decoder $p(\mathbf{X}|\mathbf{Z})$. We illustrate the graphical model of our framework in Figure 1.

### 4.1 The prior distribution

We assign a prior such that the representation $\mathbf{z}_i$ of node $i$ is a sparse vector. We use a prior that separately decides the sparse pattern and non-zero values in $\mathbf{z}_i$. Let $\mathbf{C} = (\mathbf{c}_i)_{i=1}^n$ be the random binary vectors indicating the sparse pattern, and let $\mathbf{V} = (\mathbf{v}_i)_{i=1}^n$ be the random real value vectors deciding non-zero entries of $\mathbf{Z}$. Both of them have the same dimension as $\mathbf{Z}$. Then

$$\mathbf{c}_{ik} \sim \text{Bernoulli}(\delta), \quad \mathbf{v}_{ik} \sim \text{Gaussian}(u, s), \quad \text{where } i = 1, \ldots, n; \ k = 1, \ldots, K \tag{9}$$

Here $\delta = 0.5$, $u = 0$, $s = 1$, and $\mathbf{Z} = \mathbf{C} \odot \mathbf{V}$ with $\odot$ being the Hadamard product. The prior $p(\mathbf{Z})$ in our framework is equivalent to $p(\mathbf{C}, \mathbf{V})$. We can view $\mathbf{z}_i$ as $i$-th node's membership assignment in $K$ communities. Each entry of $\mathbf{z}_i$ is from a spike and slab distribution: the spike indicates whether the $i$-th node belongs to the corresponding community, while the slab indicates the (positive or negative) strength of $i$-th node's membership in the community.

### 4.2 The conditional distribution of the adjacency

We then define the generative process of graph edges. We use the OSBM (Latouche et al., 2011) to define the decoder for $\mathbf{A}$. In OSBM, two nodes' community memberships indicate how they interact. We use parameter $\mathbf{B} \in \mathbb{R}^{K \times K}$ to indicate the interactions between different communities. A large value in the entry $B_{kk'}$ means a node in community $k$ has a higher chance of connecting with a node in community $k'$, and vice versa. Formally, we define the graph edge distribution as follows:

$$p_{\mathbf{B}}(A_{ij} = 1|\mathbf{z}_i, \mathbf{z}_j) = \text{sigmoid}(\mathbf{z}_i^\top \mathbf{B} \mathbf{z}_j), \quad p_{\mathbf{B}}(\mathbf{A}|\mathbf{Z}) = \prod_{i=1}^n \prod_{j=1}^n p_{\mathbf{B}}(A_{ij}|\mathbf{z}_i, \mathbf{z}_j). \tag{10}$$

Here $\mathbf{B}$ is learned together with other parameters.

The OSBM distribution is important for the NORAD model to encode meaningful community structures in the graph into node representations. To explain the graph structure $\mathbf{A}$, $\mathbf{z}_i^\top \mathbf{B} \mathbf{z}_j$ needs to be large for connections and small for non-edges. While the model can freely decide the $\mathbf{B}$ matrix, later experiments indicate that the learned $\mathbf{B}$ has relatively large positive diagonal elements. It means that the nodes belonging to the same communities (having positive entries corresponding to these communities) tend to have a connection in between.

### 4.3 The conditional distribution of attributes

We also define an interpretable decoder for node attributes. Here we first focus on binary attributes: $\mathbf{x}_i \in \{0, 1\}^D$ is a binary vector.

We assume that each $\mathbf{z}_i$ independently generates $\mathbf{x}_i$: $p(\mathbf{X}|\mathbf{Z}) = \prod_{i \in V} p(\mathbf{x}_i|\mathbf{z}_i)$. Then we design an Attention-based Topic Network (ATN) to consider the community-attribute relation in $p(\mathbf{x}_i|\mathbf{z}_i)$. We use $\theta$ to denote the network parameters in ATN.

Similar to topic models, ATN assumes that each community is represented as an embedding vector. All such embedding vectors are in a matrix $\mathbf{T} \in \mathbb{R}^{K \times d'}$. Each attribute also has an embedding vector, and all attributes' vectors are denoted in a matrix $\mathbf{U} \in \mathbb{R}^{d' \times D}$. We compute attention weights from the two embedding matrices to decide the probabilities of the node's attributes from the two embedding matrices. We first compute the aggregated community vector

$$\mathbf{g}_i = \text{relu}(\mathbf{T}^\top \mathbf{z}_i). \tag{11}$$

---

**Algorithm 1** Variational EM for NORAD

---

**Input:** network graph $G = (\mathbf{A}, \mathbf{X})$, initialized model and variational parameters $(\theta, \phi)$ and blockmodel $\mathbf{B}$, iteration steps $T^e, T^m$, ratio factor $\alpha$, regularization factor $\gamma$.
**Output:** learned model and variational parameters $(\theta, \phi)$ and blockmodel $\mathbf{B}$.
**repeat**
  [**E-step**]
  Fix blockmodel $\mathbf{B}$
  **for** $t = 1, \ldots, T^e$ **do**
    Compute $(\boldsymbol{\eta}, \boldsymbol{\mu}, \boldsymbol{\sigma}) = \text{gnn}(\mathbf{X}, \mathbf{A}; \phi)$.
    Sample $\mathbf{C} \sim \text{Bernoulli}(\boldsymbol{\eta})$, $\mathbf{V} \sim \text{Gaussian}(\boldsymbol{\mu}, \boldsymbol{\sigma}^2)$ via reparameterization tricks.
    Compute Hadamard product $\mathbf{Z} = \mathbf{C} \odot \mathbf{V}$.
    Compute loss $\mathcal{L}(\theta, \phi) = \alpha \log p_\theta(\mathbf{X}|\mathbf{Z}) + \log p_\mathbf{B}(\mathbf{A}|\mathbf{Z}) + \log p(\mathbf{C}, \mathbf{V}) - \log q_\phi(\mathbf{C}, \mathbf{V}|\mathbf{A}, \mathbf{X})$.
    Update $(\phi, \theta)$ by maximizing $\mathcal{L}(\phi, \theta)$ via SGD.
  **end for**
  [**M-step**]
  Fix parameters $(\phi, \theta)$
  **for** $t = 1, \ldots, T^m$ **do**
    Compute $(\boldsymbol{\eta}, \boldsymbol{\mu}, \boldsymbol{\sigma}) = \text{gnn}(\mathbf{X}, \mathbf{A}; \phi)$.
    Compute Hadamard product $\mathbf{Z} = \boldsymbol{\eta} \odot \boldsymbol{\mu}$.
    Compute loss $\mathcal{L}(\mathbf{B}) = \log p_\mathbf{B}(\mathbf{A}|\mathbf{Z}) - \gamma \|\mathbf{B}\|$.
    Update $\mathbf{B}$ by maximizing $\mathcal{L}(\mathbf{B})$ via SGD.
  **end for**
**until** convergence of the ELBO $\mathcal{L}(\phi, \theta, \mathbf{B})$

---

Then we use the vector $\mathbf{g}_i$ as a query to compute attention weights against attribute vectors $\mathbf{U}$.

$$\boldsymbol{\lambda}_i = \text{sigmoid}\left(\frac{\mathbf{g}_i^\top \mathbf{W}_q \mathbf{W}_k^\top \mathbf{U}}{\sqrt{d''}}\right). \tag{12}$$

Here the parameters $\mathbf{W}_q$ and $\mathbf{W}_k$ both have size $(d' \times d'')$. We apply the sigmoid function, instead of softmax in standard attention calculation, to get binary probabilities $\boldsymbol{\lambda}_i$.

We denote the procedure above as $\boldsymbol{\lambda}_i = \text{atn}(\mathbf{z}_i; \mathbf{T}, \mathbf{U}, \mathbf{W})$, then we have

$$p(\mathbf{x}_i|\mathbf{z}_i) = \prod_{d=1}^{D} p(\mathbf{x}_{id}|\mathbf{z}_i), \quad p(\mathbf{x}_{id} = 1|\mathbf{z}_i) = \boldsymbol{\lambda}_{id}. \tag{13}$$

The calculation in (11) and (12) capture the interaction between communities and "topics" in $\mathbf{X}$. The $\mathbf{U}$ matrix has a shape such that $d'$ is much smaller than $D$, which means that it must capture correlations among node attribute entries, and each row of $\mathbf{U}$ can be viewed as a topic. The three matrices $\mathbf{T}$, $\mathbf{W}_q$, and $\mathbf{W}_k$ roughly decides how community components in $\mathbf{z}_i$ activate these topics. Our experiment later does indicate the topic structure in $\mathbf{U}$.

For node features that are not binary, we can devise appropriate distributions and parameterize them with $\boldsymbol{\lambda}_i$.

## 4.4 The encoder

NORAD's encoder is designed for two purposes: computing node representations and model fitting through variational inference, which we will discuss right after this subsection.

The encoder $q_\phi(\mathbf{Z}|\mathbf{A}, \mathbf{X})$ is computed by a GNN, whose parameters are collectively denoted by $\phi$. Since $\mathbf{Z}$ is computed from $\mathbf{C}$ and $\mathbf{V}$, we use $q_\phi(\mathbf{C}, \mathbf{V}|\mathbf{A}, \mathbf{X})$ instead, this can be further represented as $q_\phi(\mathbf{Z}|\mathbf{A}, \mathbf{X}) = q_\phi(\mathbf{V}|\mathbf{A}, \mathbf{X})q_\phi(\mathbf{C}|\mathbf{A}, \mathbf{X})$ by using mean-field distribution. In the variational distribution, $\mathbf{V}$ and $\mathbf{C}$ are

respectively sampled from Bernoulli and Gaussian distributions, which are parameterized by a GNN.

$$q_\phi(\mathbf{C}|\mathbf{A}, \mathbf{X}) \sim \text{Bernoulli}(\boldsymbol{\eta}), \quad q_\phi(\mathbf{V}|\mathbf{A}, \mathbf{X}) \sim \text{Gaussian}(\boldsymbol{\mu}, \boldsymbol{\sigma}^2). \tag{14}$$

We use a GNN to compute these distribution parameters from $\mathbf{X}$ and $\mathbf{A}$.

$$(\boldsymbol{\eta}, \boldsymbol{\mu}, \boldsymbol{\sigma}) = \text{gnn}(\mathbf{A}, \mathbf{X}; \phi). \tag{15}$$

Here $\boldsymbol{\eta}$, $\boldsymbol{\mu}$, and $\boldsymbol{\sigma}$ are all matrices of size $n \times K$. The output of the GNN has size $n \times (3K)$, then it is split into the three matrices. All parameters of the two variational distributions are network parameters of the GNN.

Deterministic node representations $\mathbf{Z}^\circ$ are computed from $\boldsymbol{\eta}$ and $\boldsymbol{\mu}$ directly.

$$\mathbf{Z}^\circ = \boldsymbol{\mu} \odot \mathbf{1}(\boldsymbol{\eta} > 0.5). \tag{16}$$

Here $\mathbf{1}(\boldsymbol{\eta} > 0.5)$ gets a binary matrix indicating which elements are greater than the threshold 0.5.

### 4.5 Model fitting through variational inference

In this section, we discuss the learning procedure of our model. Besides the ELBO we have discussed in the previous section, we also add regularization terms over the parameter $\mathbf{B}$.

$$\mathcal{L}(\theta, \phi, \mathbf{B}) = \mathrm{E}_{q_\phi(\mathbf{C}, \mathbf{V}|\mathbf{A}, \mathbf{X})}\big[\log p_\mathbf{B}(\mathbf{A}|\mathbf{C}, \mathbf{V}) + \alpha \log p_\theta(\mathbf{X}|\mathbf{C}, \mathbf{V}) + \log p(\mathbf{C}, \mathbf{V}) - \log q_\phi(\mathbf{C}, \mathbf{V}|\mathbf{A}, \mathbf{X})\big] + \gamma\|\mathbf{B}\|. \tag{17}$$

The hyper-parameter $\alpha$ controls the strength of the attribute decoder. We maximize the objective $\mathcal{L}(\theta, \phi, \mathbf{B})$ with respect to model parameters ($\theta$ and $\mathbf{B}$) and variational parameters $\phi$. The objective gradient is estimated through Monte Carlo samples from the variational distribution. Note that the random variable $\mathbf{C}$ is binary, and we use the Gumbel-softmax trick (Jang et al., 2016), then we can estimate the gradient of the objective efficiently through the reparameterization trick.

We find alternatively updating $(\theta, \phi)$ and $\mathbf{B}$ in a variational-EM fashion gives better optimization performance. In our optimization procedure, we fix $\mathbf{B}$ and update $(\theta, \phi)$ for a few iterations at E-step, and then fix model and variational parameters $(\theta, \phi)$ and optimize $\mathbf{B}$ for a few iterations at M-step. The parameters are all optimized with SGD. The intuition behind this is that the training of $\mathbf{B}$ needs to depend on somewhat clear relations between different communities. The training procedure is shown in Algorithm 1.

### 4.6 Rectifying representations of isolated nodes

In the training process, the representations of isolated nodes are learned to predict zero connections from these nodes. As we have analyzed, representations learned in this approach are biased. Here we use the rectification strategy to post-process the learned representations of isolated nodes. For an isolated node $i$, we first compute the deterministic node representation $\mathbf{z}_i^\circ$ from Equation (16) and then update $\mathbf{z}_i^\circ$ through ATN to improve the recovery of $\mathbf{x}_i$. The update rule is shown as follows:

$$\mathbf{z}_i^\circ = \mathbf{z}_i^\circ + \epsilon\nabla_{\mathbf{z}_i^\circ}\log p_\theta(\mathbf{x}_i|\mathbf{z}_i^\circ), \tag{18}$$

where $\epsilon$ is the updated learning rate. We run the update for multiple iterations and obtain the final representation. Empirically, 50 to 100 iterations usually give a clear improvement of these nodes' representations.

## 5 Experiments

In this section, we study the proposed model with real datasets. The first aim of the study is to examine the quality of node representations: whether the model learns node representations of high quality and how each component contributes to the learning. We examine our model through extensive ablation studies and sensitivity analysis. The second aim is to examine the interpretability of learned node representations. We look into the data and show how learned representations encode interpretable structures in the data.

|  |  | DeepWalk | VGAE | KernelGCN | DGLFRM | VGNAE | NORAD |
|---|---|---|---|---|---|---|---|
| Cora | AUC | 84.6±0.01 | 92.6±0.01 | 93.1±0.06 | 93.4±0.23 | 94.9±0.43 | **95.6±0.56** |
|  | AP | 88.5±0.00 | 93.3±0.01 | 93.2±0.07 | 93.8±0.22 | 94.9±0.39 | **96.1±0.44** |
| Citeseer | AUC | 80.5±0.01 | 90.8±0.02 | 90.9±0.08 | 93.8±0.32 | **96.0±0.74** | 95.6±0.28 |
|  | AP | 85.0±1.00 | 92.0±0.02 | 91.8±0.04 | 94.4±0.73 | **96.1±0.89** | 96.5±0.19 |
| Pubmed | AUC | 84.2±0.02 | 94.2±0.76 | 94.5±0.03 | 94.0±0.08 | 95.0±0.26 | **97.1±0.25** |
|  | AP | 87.8±1.00 | 94.0±0.88 | 94.2±0.01 | 95.0±0.35 | 94.7±0.36 | **97.3±0.28** |
| DBLP | AUC | 80.4±0.65 | 90.8±0.37 | 93.6±0.22 | 93.7±0.41 | 91.8±0.34 | **96.3±0.22** |
|  | AP | 83.1±0.55 | 91.4±0.44 | 93.9±0.18 | 94.0±0.54 | 92.6±0.26 | **97.0±0.18** |
| OGBN-arxiv | AUC | OOM | 87.2±1.37 | OOM | 91.2±0.38 | **92.3±5.03** | 94.5±0.22 |
|  | AP | OOM | 88.7±1.17 | OOM | 92.0±0.31 | **92.8±4.12** | 94.6±0.20 |
| Wiki-cs | AUC | OOM | 92.1±0.99 | 93.1±0.85 | 92.8±0.90 | 95.3±0.43 | **96.0±0.22** |
|  | AP | OOM | 91.5±0.94 | 93.0±1.22 | 93.0±1.07 | 94.5±0.55 | **96.0±0.23** |

Table 1: Performance comparison of all models in the link prediction task: We use unpaired t-test to compare models' performance values. Not significantly worse than the best at the 5% significance level are bold.

|  |  | DeepWalk | VGAE | KernelGCN | DGLFRM | VGNAE | NORAD |
|---|---|---|---|---|---|---|---|
| Cora | NMI | 40.0±1.26 | 42.6±2.64 | 44.2±1.07 | 48.0±2.02 | **51.1±0.84** | 50.3±3.72 |
|  | ACC | 56.5±1.71 | 56.7±4.49 | 60.9±2.43 | 63.1±4.11 | **67.5±2.29** | 66.5±5.89 |
| Citeseer | NMI | 13.2±1.55 | 15.5±2.83 | 25.6±2.56 | 28.8±1.63 | 35.6±3.76 | **38.9±1.18** |
|  | ACC | 38.8±2.12 | 36.1±2.38 | 52.8±4.36 | 51.9±2.38 | 57.8±4.41 | **64.5±1.17** |
| Pubmed | NMI | **28.5±0.73** | **30.1±2.56** | **28.6±1.27** | 25.0±4.21 | 26.2±0.47 | 24.5±5.97 |
|  | ACC | 67.1±0.54 | **67.6±2.88** | **68.6±0.82** | 65.2±3.64 | 64.1±0.37 | 61.1±6.36 |
| DBLP | NMI | 19.7±1.76 | 23.6±2.70 | 30.5±0.56 | 30.0±2.66 | 26.0±3.40 | **40.2±4.59** |
|  | ACC | 54.8±1.31 | 47.8±2.81 | 56.0±2.61 | 55.8±3.19 | 52.4±6.66 | **64.4±6.09** |
| OGBN-arxiv | NMI | OOM | 7.3±2.15 | OOM | **9.4±2.55** | 9.0±2.15 | **10.3±1.64** |
|  | ACC | OOM | **10.3±1.15** | OOM | **11.2±0.89** | 10.4±0.97 | **11.1±0.69** |
| Wiki-cs | NMI | OOM | 21.2±5.66 | **34.1±5.20** | 20.4±6.69 | 24.6±1.10 | **34.1±1.57** |
|  | ACC | OOM | 29.9±3.63 | **39.1±3.86** | 30.4±4.33 | 33.7±1.01 | **39.4±2.24** |

Table 2: Performance comparison of all models in the node clustering task: We use unpaired t-test to compare models' performance values. Not significantly worse than the best at the 5% significance level are bold.

**Datasets.** We use seven benchmark datasets, including Cora, Citeseer, Pubmed, DBLP, OGBL-collab, OGBN-arxiv, and Wiki-cs (Morris et al., 2020; Hu et al., 2020; Mernyei & Cangea, 2020). The details of these datasets can be found in Table 8 in Appendix.

**Baselines.** We benchmark the performance of NORAD against existing models that share the same traits as ours. The experiment setups and the implementation details of our model are shown in Appendix A.2. We consider five baseline methods. (1) DeepWalk (Perozzi et al., 2014) learns the latent node representations by treating truncated random walks sampled within the network as sentences. (2) VGAE (Kipf & Welling, 2016) is the first variational auto-encoder based on GNN. (3) KernelGCN (Tian et al., 2019) proposes a learnable kernel-based framework, which decouples the kernel function and feature mapping function in the propagation of GCN; (4) DGLFRM (Mehta et al., 2019) adopts IBP and Normal prior when encoding the node representations and adds a perceptron layer to the node representations before decoding the edges. (5) VGNAE (Ahn & Kim, 2021) shows that L2 normalization on node hidden vector in GCN improves the representation quality of isolated nodes.

## 5.1 Link prediction and node clustering

We consider the quality of node representations in link prediction and node clustering tasks. In particular, we pay attention to the representations of isolated nodes in sparse graphs. In the link prediction task, all

|            | VGAE             | KernelGCN | DGLFRM          | VGNAE           | NORAD           |
|------------|------------------|-----------|-----------------|-----------------|-----------------|
| Hits@10↑   | 26.98±0.12       | OOM       | 28.75 ±0.68     | 31.24±0.19      | **31.56±0.23**  |
| Hits@50↑   | 44.76±0.98       | OOM       | 46.19±1.37      | **46.76±0.88**  | **47.38±1.09**  |
| Hits@100↑  | **51.44±0.70**   | OOM       | **51.49±0.15**  | 51.39±0.42      | 51.40±0.25      |

Table 3: Performance comparison of all models on OGBL-collab dataset: We use unpaired t-test to compare models' performance values. Not significantly worse than the best at the 5% significance level are bold.

models predict the missing edges of the graph based on the training data, including node attributes and links. We evaluate our model and the baselines regarding Area Under the ROC Curve (AUC) and Average Precision(AP) on six datasets. We follow the data splitting strategy in Kipf & Welling (2016) and report the mean and standard deviation over 10 random data splits. For Wiki-cs, since all models yield high performance by following the data splitting strategy in VGAE as mentioned in (Mernyei & Cangea, 2020), we lower down the training ratio and use split ratio: train/val/test: 0.6/0.1/0.3. The results are reported in Table 1. Note that an "OOM" entry means that the corresponding model cannot run on that dataset because of the out-of-memory issue. Since the results on OGBL-collab use different metrics, we report the metric Hits@K, K=10, 50, 100 following (Hu et al., 2020) in Table 3. Given that KernelGCN requires transformation on the adjacency matrix, the computation memory cost is too high to scale it to large networks, e.g., OGBN-arxiv and OGBL-collab.

The table shows that NORAD significantly outperforms the baselines on almost all datasets, illustrating the superior capability of our model in encoding graph information. The advantage is more obvious on DBLP, which has relatively rich node attributes and links.

In the node clustering task, we generate the node representations from different learning models and then use K-means to obtain the clustering results. We set the hyperparameter in K-means as the number of classes in each dataset. We compare the clustering performance with the node embeddings learned from the baselines above. We compare NORAD against the baselines in terms of Normalized Mutual Information (NMI) and Accuracy (ACC) on six datasets. Before calculating ACC, we use the Hungarian matching algorithm (Kuhn, 1955) to match the K-means predicted cluster labels with true labels. We report the mean and standard derivation over 10 runs. The results can be found in Table 2.

In the node clustering task, NORAD works better on Cora, Citeseer, DBLP, OGBN-arxiv, and Wiki-cs against almost all baselines. For Citeseer and DBLP, NORAD significantly outperforms all five baselines; this can be explained by the fact that these two datasets provide more node information or edge information than the other two. Our performance on Pubmed is not as competitive as on other datasets. One possible reason is that Pubmed has the least node information among all the six datasets, and node attributes are not informative about node classes.

## 5.2 Ablation study

In this section, we do ablation studies to understand the benefit of each model's component in the link prediction task. Specifically, we investigate different variants of the decoder for $\mathbf{A}$ and $\mathbf{X}$, respectively, and the benefits of employing representation rectification for isolated nodes. The results are tabulated in Table 4, and implementation details can be found in Appendix A.2.

We compare our edge decoder against two variants, whose results are shown in the second and third rows in Table 4. The first variant constructs the decoder by $p(\mathbf{A}, \mathbf{X}|\mathbf{Z}) = p(\mathbf{X}|\mathbf{Z})p(\mathbf{A}|\mathbf{X}, \mathbf{Z})$, that is, we first decode $\mathbf{X}$ from $\mathbf{Z}$ and then decode $\mathbf{A}$ from $\mathbf{Z}$ and decoded $\mathbf{X}$. We find that this model is less capable of explaining the adjacency matrix because it mainly encodes information of node features. The decoding of the adjacency matrix depends on decoded features $\mathbf{X}$, whose dimension is much higher than $\mathbf{Z}$ and likely to cause extra errors. The second variant uses the dot product decoder $p(\mathbf{A}|\mathbf{Z})$ from the VGAE model. It essentially uses an identity matrix as $\mathbf{B}$. The dot product decoder in the second variant shows much worse performance than our decoder. This result shows the benefit of learning the block model $\mathbf{B}$, which can control connection densities within/between the same/different communities.

| Model Variants | Cora | Citeseer |
|---|---|---|
| NORAD | 95.6±0.56 | 95.6±0.28 |
| construction w/ $p(\mathbf{A}|\mathbf{X}, \mathbf{Z})$ | 94.5±0.42 | 94.3±0.76 |
| VGAE decoder $p(\mathbf{A}|\mathbf{Z})$ | 90.4±1.21 | 89.9±1.01 |
| w/o decoder $p(\mathbf{X}|\mathbf{Z})$ | 94.6±0.58 | 93.4±0.99 |
| NRTM decoder $p(\mathbf{X}|\mathbf{Z})$ | 95.3±0.41 | 94.0±0.42 |
| TAN decoder $p(\mathbf{X}|\mathbf{Z})$ | 95.0±0.69 | 94.7±0.55 |
| w/o rectification | 95.0±0.62 | 95.0±0.29 |

Table 4: AUC of link prediction on Cora and Citeseer using different model variants.

We then investigate the power of using different node decoders $p(\mathbf{X}|\mathbf{Z})$. Specifically, we choose decoders proposed by Neural Relational Topic Model (NRTM) (Bai et al., 2018) and Topic Attention Networks (TAN) (Panwar et al., 2021). In addition, the effect of the node decoder absence is also considered. We find that our ATN decoder performs better than the decoders in other works. Moreover, we can see that using the node decoder significantly contributes to model performance. Interestingly, the richer the node attributes are, the more the model can benefit from utilizing a node decoder.

Finally, we study how rectifying the representations of the isolated nodes brings additional performance rise to our model. Even though only a small portion of the nodes are isolated, we still observe a non-trial improvement in the performance, which indicates the effectiveness of rectification. We give a more thorough analysis of it under the circumstance when graphs are sparse in the next section,

## 5.3   Attribute decoding for sparse graphs

Our experiments show strong evidence that decoding attribute benefits node representation learning when graphs are sparse. We create sparser graphs by masking out more edges during training. Specifically, we use different training ratios (TRs) and keep 20%, 40%, 60%, and 80% edges in the training set. For the remaining edges, 1/3 are used for validation, and 2/3 are used for testing. We again choose Cora and Citeseer datasets for analysis. Results are reported in Table 5. This result shows that more benefit is gained from the node decoder ATN when graphs are sparser.

We then consider the case of strengthening the node decoder by using $\alpha$ values greater than 1. We run the same experiment as above but varying $\alpha$ values in Equation (17) and then report our observations in Table 6. We see that large $\alpha$ values slightly increase the performance for sparse graphs.

Suggesting links for isolated nodes is often considered as the cold-start problem (Schein et al., 2002), which is important in recommender systems. Here we examine the benefit of the rectification procedure proposed in Equation (18) by checking the isolated node representations. We consider datasets with different sparsity. A lower training ratio indicates more isolated nodes exist. The details of isolated nodes for each dataset can be found in Table 9 in Appendix. The results in Figure 2 show that our rectification procedure greatly improves the quality of isolated nodes' representations.

## 5.4   Interpretability of node representations

We inspect node representations learned by our model qualitatively and interpret them in the application context. Specifically, we focus on two questions: whether representations capture community structures in the data and whether representations explain node attributes from the perspective of community structures.

**Alignment between community membership and node classes.**   We visualize a few example communities detected from the graph and correspond them to node labels. The visualization is through the following steps. (1) We first obtain node representations $\mathbf{Z}$ from a trained model and compress them into two-dimensional vectors by using t-SNE (Van der Maaten & Hinton, 2008). (2) Then we take 10% of the nodes and choose 2 representative communities (corresponding to two entries in a node vector). Note that we perform the dimension reduction in full data, then randomly take a subset of the compressed data. (3) In the

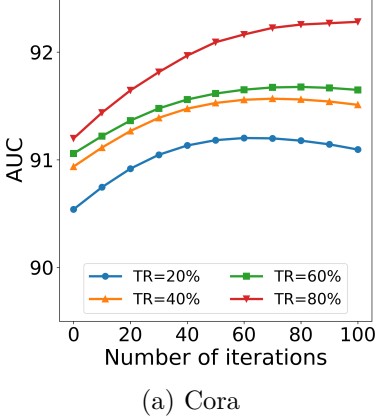

(a) Cora

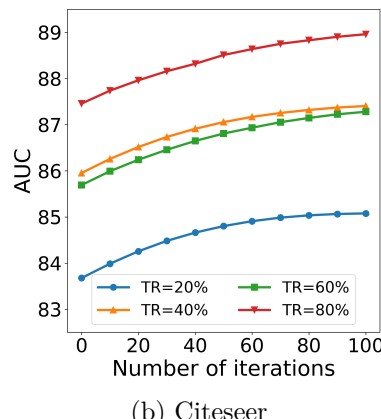

(b) Citeseer

Figure 2: Rectifying representations of isolated nodes: for both (a) Cora and (b) Citeseer datasets, we can find that the AUC of link prediction related to isolated nodes is increasing with the rectification iteration on all training ratios (TRs).

| Ratio | Cora | | Citeseer | |
|---|---|---|---|---|
| | w/o ATN | w/ ATN | w/o ATN | w/ ATN |
| 20% | 82.7±0.61 | 86.4±0.58 | 88.5±0.76 | 92.2±0.55 |
| 40% | 88.5±0.71 | 90.7±0.44 | 91.7±0.51 | 93.7±0.39 |
| 60% | 91.9±0.63 | 93.2±0.37 | 92.8±0.58 | 94.5±0.42 |
| 80% | 94.0±0.75 | 95.2±0.56 | 93.9±0.62 | 95.6±0.48 |

Table 5: Benefits of using ATN versus different sparsity levels: The percentage(%) indicates the fraction of graph edges in the training set. We report the link prediction performance (AUC).

| $\alpha$ | Cora | | Citeseer | |
|---|---|---|---|---|
| | 20% | 40% | 20% | 40% |
| 1.0 | 86.4 | 90.7 | 92.2 | 93.7 |
| 1.5 | 86.5 | 90.8 | 92.6 | 93.9 |
| 2.0 | 86.7 | 90.9 | 92.8 | 94.0 |
| 2.5 | 86.8 | 91.1 | 92.9 | 94.1 |
| 3.0 | 86.9 | 91.4 | 93.1 | 94.3 |

Table 6: Link prediction performance (AUC) on Cora and Citeseer with different training ratios as $\alpha$ increases.

last step, we set the threshold to 0.5 and compute the community membership for each selected node. There are three kinds of community assignments for each node – community 1, community 2, and others.

The visualization results are shown in Figure 3. We use colors to indicate node classes. Then we extract communities from $\mathbf{Z}$. For community $k$, nodes with $\mathbf{z}_{ik}$ greater than a threshold are considered to be in the community. We identify nodes in two communities and indicate them with markers in the figure. We see that nodes in the same community tend to be in the same node class. They also tend to clump together in the plot.

**Topics used by communities.** Here, we inspect the relation between learned communities and node attributes by analyzing node representations learned from the Pubmed dataset. In this dataset, each node is a clinical article, the node attribute is a binary vector indicating words appearing in this article, and links represent their citation relations. The articles in Pubmed are categorized into three Diabetes Mellitus classes: (1) Experimental, (2) Type 1, and (3) Type 2. We examine whether nodes in the same community share similar topics. To determine which words are used by each community $k$, we manually set a $\mathbf{c}$ to be a one-hot vector with $\mathbf{c}_k = 1$ and $\mathbf{v} \sim \text{Gaussian}(\mathbf{0}, \mathbf{I})$, then we get a fake node vector $\mathbf{z} = \mathbf{c} \odot \mathbf{v}$ from community $k$. We sample $\mathbf{z}$ 10,000 times, reconstruct $\mathbf{x}$ via ATN, and get a distribution of words ("topics") used by community $k$. These words do not include non-semantic words, which are removed by thresholding word frequency.

Our model learned 12 related subareas, which are highly related to their own corresponding high-frequency stemmed keywords. In the appendix, we show the full names of the abbreviations in Table 10. We show each community's top eight stemmed keywords in Table 7. For each community, these keywords are coherently related to a medical subarea, such as treatment, disease, etc. Here we manually assign each community a label according to the area. For example, by inspecting the keywords, the second community can be labeled

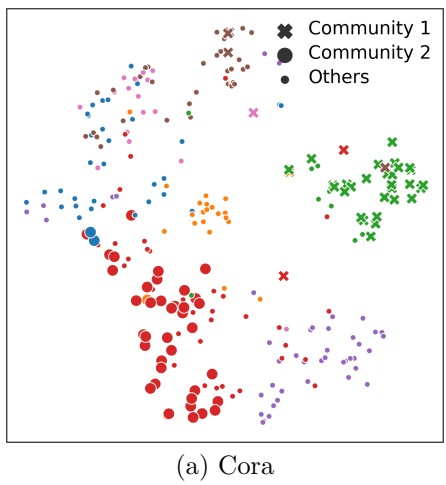
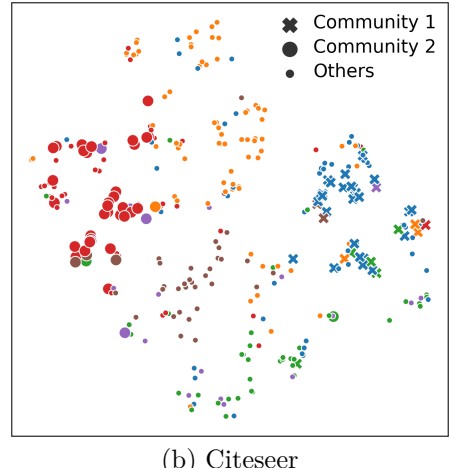

(a) Cora

(b) Citeseer

Figure 3: The t-SNE visualization of node representations on Cora and Citeseer: color indicates the node class, marker indicates the community assignment. We only choose two non-overlapping communities for visualization.

| Topic Label | Top Eight Stemmed Keywords |
| --- | --- |
| NAFLD | liver, fat, resist, correl, hepat, befor, obes, degre |
| DR | retinopathi, complic, diagnosi, mortal, famili, cohort, predict, screen |
| CAD | cholesterol, genotyp, polymorph, lipoprotein, heart, coronari, target, triglycerid |
| Hypoglycemic agents | exercis, postprandi, oral, presenc, region, metformin, reduct, agent |
| Diet intervention | intervent, intak, particip, promot, loss, individu, primari, dietari |
| Eating disorder | depress, chronic, approxim, dietari, secret, cpeptid, defect, vs |
| Obesity | period, pathogenesi, greater, continu, daili, meal, weight, dose |
| AGEs | receptor, alter, neuropathi, streptozotocin, oxid, inhibit, stimul, peripher |
| ICA | antibodi, transplant, beta, pancreat, antigen, ica, immun, tcell |
| HbA1c | hypoglycaemia, hba1c, manag, symptom, life, glycaem, known, ii |
| MS neuropathy | abnorm, metabol, children, nondiabet, durat, nerv, rat, sever |
| MS uric acid | caus, syndrom, evid, acid, diet, urinari, rat, insulindepend |

Table 7: Display of top eight words in twelve learned latent topics in Pubmed dataset. The topic labels are abbreviations of clinical subareas related to Diabetes Mellitus. They are highly related to the top eight stemmed keywords. The details can be found in Table 10 in Appendix.

as diabetic retinopathy (DR), which is the most common complication of diabetes mellitus (Wang & Lo, 2018); the third community can be labeled as coronary artery disease (CAD), which is happened at a higher risk in patients with type 2 diabetes mellitus (T2DM) than non-T2DM patients Naito & Kasai (2015); the ninth community can be labeled as Islet Cell Antibodies (ICA)[*].

It shows that the learned subareas of diabetes mellitus are all meaningful and interpretable after analyzing high-frequency stemmed keywords. Besides, We also find that each community often belongs to one node class in the dataset, which again proves that the detected community is coherent.

---

[*]According to Narendran et al. (2005): "the **beta** cell: **ICA**, is the first islet (a **pancreatic** cell) '**autoantigen**' to be described. **Antibodies** to **ICA** are present in 90% of **type 1** diabetes patients at the time of diagnosis."

# 6 Related Work

The stochastic blockmodel (Wang & Wong, 1987; Snijders & Nowicki, 1997) is frequently used for detecting and modeling community structure within network data. Later variants of SBM make different assumptions on the node-community relationships (Airoldi et al., 2008; Miller et al., 2009; Latouche et al., 2011). For example, the mixed membership stochastic blockmodel (MMSB) (Airoldi et al., 2008) assumes each node belongs to a mixture of communities. The overlapping stochastic blockmodel (OSBM) (Latouche et al., 2011) assumes each node can belong to multiple communities with the same strength.

Variational framework (Blei et al., 2017) are frequently applied in learning graph data (Kipf & Welling, 2016; Chen et al., 2021). The VGAE models (Kipf & Welling, 2016; Hasanzadeh et al., 2019; Mehta et al., 2019; Sarkar et al., 2020; Li et al., 2020; Cheng et al., 2021) combine a VAE and a GNN to learn the representation of graph data. DGLFRM (Mehta et al., 2019) replaces the Gaussian prior with Indian Buffet Process (IBP) prior (Teh et al., 2007) to promote the interpretability of learned representations. LGVG (Sarkar et al., 2020) extends the ladder VAE (Sønderby et al., 2016) to modeling graph data and introduces the gamma distribution to enable the interpretability of the learned representation. But these models do not explain the relationship between communities and attributes. Other models improve the architectures of different components of VGAE. DGVAE (Li et al., 2020) instead uses the Dirichlet prior and shows its application in node clustering and balanced graph cut. Cheng et al. (2021) devises a model that decodes multi-view node attributes.

The RTM models (Nallapati et al., 2008; Chang & Blei, 2009) focus on learning meaningful topics from the document content with the help of the relational information that resides in the document network. Various works (Wang et al., 2017; Bai et al., 2018; Xie et al., 2021; Panwar et al., 2021) are proposed based on this idea by either improving the graphical model or proposing a novel network architecture.

On combining the ideas of VGAEs and RTMs, our model can learn node representations that not only being useful for performing downstream tasks such as link prediction and node clustering but also provide high interpretability in both community-wise and topic-wise, where the former concerns the topological structure of the network and the latter concerns the content of the documents.

# 7 Conclusion

In this work, we have theoretically analyzed the role of node attribute decoder in representation learning with VGAEs. We show that the node attribute decoder helps the model encode information about the graph structure.

We further propose the NORAD model to learn interpretable node representations. We introduce the OSBM to the edge decoder with the aim of capturing community structure. We carefully designed the ATN to decode node attributes, which improves the quality of node representations and makes them interpretable. We also design a rectification procedure to refine representations of isolated nodes in the graph after model training.

However, our model has the following limitations: (1) It needs substantial work to extend our node attribute decoder for other types of features. (2) Although the block model improves the performance of edge decoding, the learned matrix $\mathbf{B}$, which is desinged for capturing interactions between communities, still lacks enough interpretability.

### Acknowledgments

We thank all reviewers and editors for their insightful comments. Li-Ping Liu was supported by NSF 1908617. Xiaohui Chen was supported by Tufts RA support.

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

| Dataset | #Nodes | #Edges | #Words/#Embedding | #Classes |
|---|---|---|---|---|
| Cora | 2,708 | 5,429 | 1,433 | 7 |
| Citeseer | 3,312 | 4,732 | 3,703 | 6 |
| Pubmed | 19,717 | 44,338 | 500 | 3 |
| DBLP | 17,716 | 105,734 | 1,639 | 4 |
| OGBL-collab | 235,868 | 1,285,465 | 128 | - |
| OGBN-arxiv | 169,343 | 1,166,243 | 128 | 40 |
| Wiki-cs | 11,701 | 216,123 | 300 | 10 |

Table 8: Dataset statistics

## A  Appendix

### A.1  Datasets details

**Cora.**  Citation network consists of 2,708 documents from seven categories. The dataset contains bag-of-words feature vectors of length 1,433. The network has 5,278 links.

**Citeseer.**  Citation network consists of 3,312 scientific publications from six categories. The dataset contains bag-of-words feature vectors of length 3,703. The network has 4,732 links.

**Pubmed.**  Citation network consists of 19,717 scientific publications from three categories. The dataset contains bag-of-words feature vectors of length 500. The network has 44,338 links.

**DBLP.**  Citation network consists of 17,716 papers from four categories. The dataset contains bag-of-words feature vectors of length 1,639. The network has 105,734 links.

**OGBN-collab.**  Collaboration network consists of 235,868 authors. The dataset contains comes with a 128-dimensional feature vector obtained by averaging the word embeddings of papers published by the authors. The network has 1,285,465 links.

**OGBN-arxiv.**  Citation network consists of 169,343 papers from fourty categories. The dataset contains comes with a 128-dimensional feature vector obtained by averaging the embeddings of words in its title and abstract. The network has 1,166,234 links.

**Wiki-cs.**  Wikipedia-based network consists of 11,701 subjects from ten categories. The dataset contains node feature vectors of length 300. The network has 216,123 links.

**Isolated nodes in different training ratios.**  In Table 9, we calculate the numbers and percentages of isolated nodes in different training set split ratios, we also get the numbers and percentages of edges that contribute to generating the isolated nodes.

### A.2  Model implementation and experimental details

In this section, we introduce the NORAD implementation and some experiments' detailed setups.

**Hyperparameters setting.**  For the encoder of NORAD, we choose 1-layer Graph Convolution Network (GCN) as our encoder. Since we need to output three sets of variational parameters, we use three GCN layers separately. Each encoder shares the same output dimension. We search the number of cluster $K$ over $\{32, 64, 128, 256\}$ and find that our model is insensitive to $K$. When $K$ becomes larger (usually 64 and above), the model gives a relatively stable performance. We choose $K = 256$ for all models in the reported experiments. For other VGAE baselines, we observe a slight performance drop when increasing the dimension of the hidden layers of the encoder. Though node representations learned by NORAD are in a higher dimension, we argue that the actual number of effective entries is usually much smaller. For

| Training ratio | Cora | | | | Citeseer | | | |
|---|---|---|---|---|---|---|---|---|
| | 20% | 40% | 60% | 80% | 20% | 40% | 60% | 80% |
| # isolated nodes | 1,370 | 702 | 336 | 120 | 2,045 | 1,275 | 737 | 341 |
| % isolated nodes | 50.6% | 25.9% | 12.4% | 4.4% | 61.8% | 38.5% | 22.3% | 10.3% |
| #contributed edges | 3,584 | 1,445 | 547 | 155 | 3,883 | 1,955 | 958 | 385 |
| %contributed edges | 66.0% | 26.6% | 10.1% | 2.9% | 82.1% | 41.3% | 20.3% | 8.1% |
| Training ratio | Pubmed | | | | DBLP | | | |
| | 20% | 40% | 60% | 80% | 20% | 40% | 60% | 80% |
| #isolated nodes | 11,220 | 7,196 | 4,290 | 1,964 | 8,290 | 4,664 | 2,628 | 1102 |
| %isolated nodes | 56.9% | 36.5% | 21.8% | 10.0% | 46.8% | 26.3% | 14.8% | 6.2% |
| #contributed edges | 19,919 | 9,938 | 5,156 | 2,125 | 19,915 | 7,765 | 3,525 | 1249 |
| %contributed edges | 45.0% | 22.4% | 11.6% | 4.8% | 18.8% | 7.3% | 3.3% | 1.2% |

Table 9: Isolated nodes numbers and percentages in different training ratios.

| Abbreviation | Full Name |
|---|---|
| NAFLD | Nonalcoholic Fatty Liver Disease |
| DR | Diabetic Retinopathy |
| CAD | Coronary Artery Disease |
| AGEs | Advanced Glycation End products |
| ICA | Islet Cell Antibodies |
| MS | Metabolic Syndrome |
| AIH | Autoimmune Hepatitis |
| DN | Diabetic Neuropathy |
| ED | Erectile Dysfunction |
| TyG | Triglyceride |
| BMI | Body mass index |
| UTI | Urinary Tract Infection |
| FSD | Female sexual dysfunction |
| HGB | Hemoglobin |

Table 10: Abbreviation and the full name of learned topics.

example, we only observe 12 effective entries in the Pubmed dataset. Detailed configurations of the encoders of the baselines and our models are shown in Table 11. For the node decoder ATN, we search $(d', d'')$ over $\{(128, 64), (64, 32)\}$. We set $d'$ to 128 and $d''$ to 64 in our experiment.

**Training and prediction.** We alternatively optimize $(\phi, \theta)$ and $\mathbf{B}$. We first update $(\phi, \theta)$ for $T^e$ steps, then update $\mathbf{B}$ for $T^m$ steps, and alternate until convergence. In the implementation, we set $T^e = 10$ and $T^m = 10$. We use Adam optimizers Kingma & Ba (2014) with a learning rate of 0.001. Since we use the Gumbel-softmax Jang et al. (2016) to relax the binary vector $\mathbf{C}$, in the training process, we use temperature annealing with 0.5 to be the minimum temperature. We use the relaxed binary vector for optimizing $(\phi, \theta)$, and for optimizing $\mathbf{B}$ and prediction phase, we use the binary vector by truncating the Bernoulli parameters $\boldsymbol{\eta}$. For isolated node rectification, we use the same learning rate to rectify the representation of the isolated nodes. We optimize the representation for multiple iterations and choose the number of iterations to be 50 when reporting the experiment results.

| Hyperparameters | VGAE | KernelGCN | DGLFRM | VGNAE | NORAD |
|---|---|---|---|---|---|
| Layer type | GCN | GCN | GCN | APPNP | GCN |
| #Layers | 2 | 2 | 2 | 1 | 1 |
| Hidden dimension | {32, 16} | {32, 16} | {256, 50} | {128} | {256} |
| Prior | Gaussian | Gaussian | Gaussian+IBP | Gaussian | Gaussian+Bernoulli |

Table 11: Encoder configuration of each model

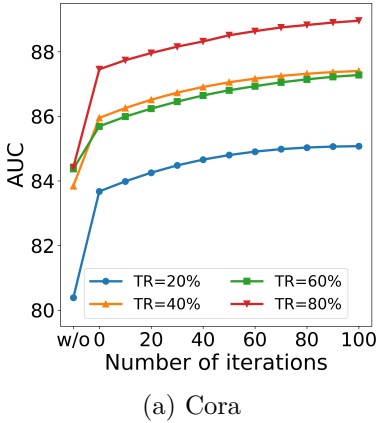
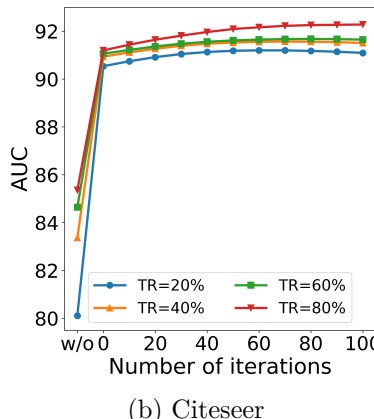

(a) Cora

(b) Citeseer

Figure 4: Isolated nodes link prediction and representation rectification: (a) For Cora, AUC of isolated nodes related link prediction is improved by adding ATN decoder (w/o → iteration 0) and increasing rectification iterations (iteration 0 → 100); (b) For Citeseer, the increase is more obvious with lower training ratio.

**Model diagnosis.** Here we show the detailed implementation for each extra component mentioned in Section 5.2. For the probability construction $p(\mathbf{Z})p(\mathbf{X}|\mathbf{Z})p(\mathbf{A}|\mathbf{Z},\mathbf{X})$, $\mathbf{Z}$ is used for two purposes: reconstructs $\mathbf{X}$ and reconstructs $\mathbf{A}$ with the help of $\mathbf{X}$. For $p(\mathbf{X}|\mathbf{Z})$, we use the ATN as the node decoder. For $P(\mathbf{A}|\mathbf{Z},\mathbf{X})$, we first use a node encoder to encode $\mathbf{X}$ into a hidden vector $\mathbf{R}$, which has the same dimension as $\mathbf{Z}$. Then we concatenate $\mathbf{Z}$ with $\mathbf{R}$ and feed it into the edge decoder. We use a two-layer MLP with a ReLU activation function for the node encoder. For the edge decoder in DGLFRM, we use one linear layer along with the ReLU activation function. We set the dimension of the output $\hat{\mathbf{Z}}$ of the MLP to be half the number of the clusters $K$. We use $\hat{\mathbf{Z}}$ to reconstruct $\mathbf{A}$ via the inner product. And for reconstructing $\mathbf{X}$, we still use $\mathbf{Z}$. Isolated node rectification is employed for all the model variants with node decoders.

### A.3 Additional Experiments

**Further result analysis of isolated nodes.** Here we show how a node decoder can greatly improve the link prediction performance for isolated nodes. Figure 4 shows the superior capability of suggesting links for isolated nodes, especially in Citeseer.

**Normalization trick on NORAD.** We also experiment with deploying normalization trick (Ahn & Kim, 2021) on our model. We add L2 normalization on the encoded features in the GCN layer. We compare the performance of adding only the normalization trick or only our ATN on the Cora and Citeseer dataset with four different ratios. We also try adding the normalization trick and our ATN to study their effects thoroughly. We show the performances of four variants in Table 12. We can find that our ATN decoder outperforms the normalization trick on both datasets with all training ratios. The difference is more obvious on the sparser graph dataset Citeseer. We also find that the combination of these two operations yields trivial improvement on Cora with some training ratios and even gets worse on Citeseer compared with using only one of the operations.

**t-SNE visualization of baselines.** In Figure 5, we visualize the node embeddings learned from NORAD, KernelGCN, DGLFRM, and VGNAE on Cora and Citeseer. Compared with other baselines, we can better interpret the community structure from the node embeddings learned by our model.

**Topic analysis for DGLFRM.** We perform the same topic analysis on the Pubmed dataset for the baseline DGLFRM. Table 13 shows the analyzed results. As we can observe, community topics learned by NORAD are more interpretable than those by DGLFRM. From a learned DGLFRM, we see that communities

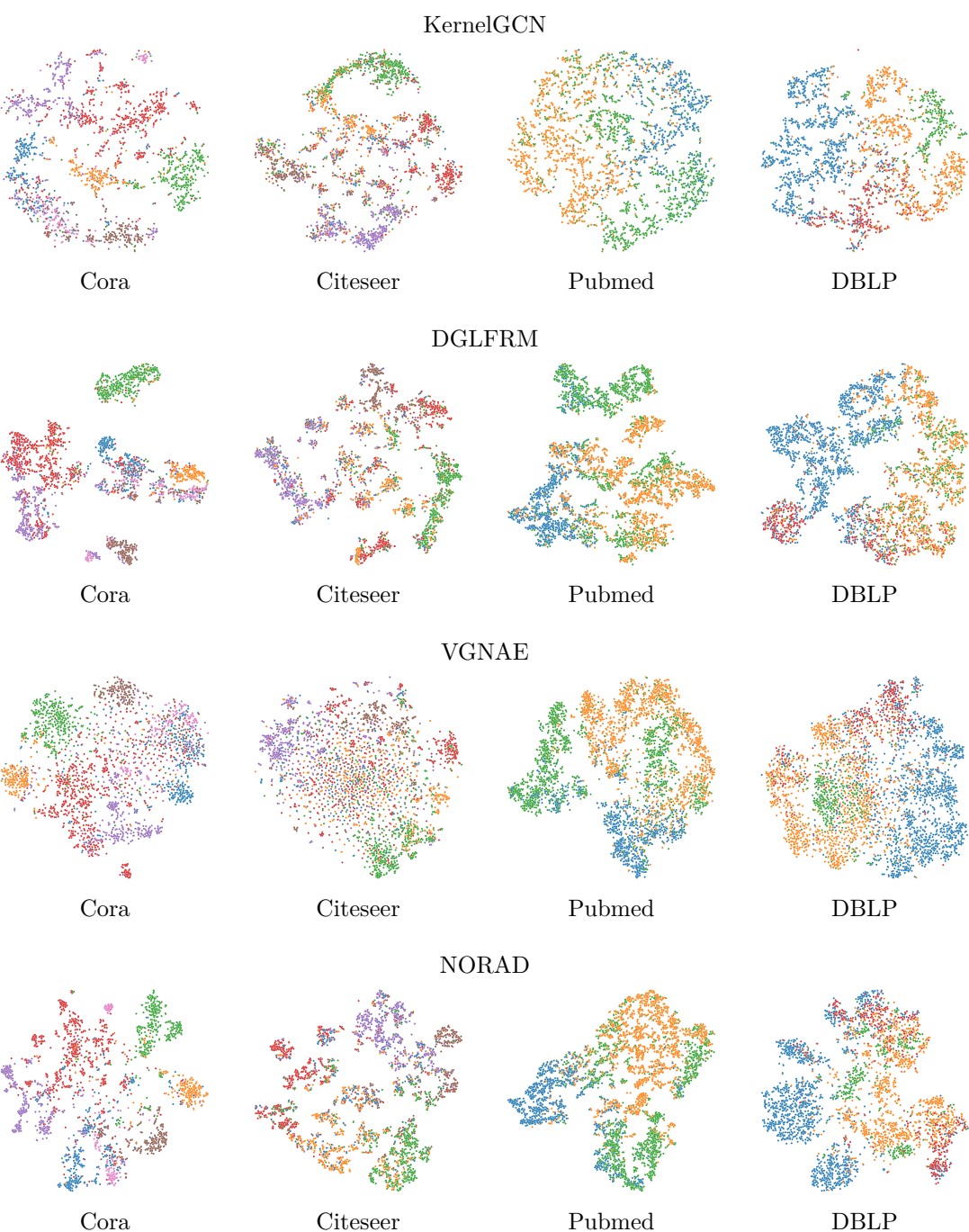

Figure 5: Visualization of latent node embeddings learned by four models on four datasets.

often have overlapped topics. As a comparison, community topics learned by NORAD are more coherent and representative.

**Downstream tasks on learned node representation.** We train a classification model for the learned node representation to see if the model has informative encoding from the data. Specifically, the classifier we adopt here is SVM with RBF kernel. The classification results are reported using Accuracy (ACC). Table 14

| Ratio | base | w/ norm | w/ ATN | w/ both |
|---|---|---|---|---|
| | | Cora | | |
| 20% | 82.7±0.61 | 86.2±1.15 | 86.4±0.58 | 87.9±0.49 |
| 40% | 88.5±0.71 | 89.8±0.52 | 90.7±0.44 | 91.2±0.51 |
| 60% | 91.9±0.63 | 91.8±0.61 | 93.2±0.37 | 92.6±0.71 |
| 80% | 94.0±0.75 | 93.0±1.25 | 95.2±0.56 | 93.7±1.14 |
| | | Citeseer | | |
| 20% | 88.5±0.76 | 85.9±1.34 | 92.2±0.55 | 91.2±0.47 |
| 40% | 91.7±0.51 | 90.1±1.04 | 93.7±0.39 | 93.3±0.72 |
| 60% | 92.8±0.58 | 91.6±1.07 | 94.5±0.42 | 94.1±0.66 |
| 80% | 93.9±0.62 | 92.8±1.09 | 95.6±0.48 | 94.8±0.70 |

Table 12: Performance comparison of normalization trick and ATN decoder: NORAD without ATN (denoted as "base"), base with normalization trick (denoted as "norm"), NORAD (denoted as ATN), NORAD with normalization trick (denoted as "both").

| Topic Label | Top Eight Stemmed Keywords |
|---|---|
| AIH | vivo, liver, releas, defect, hepat, given, content, autoimmun |
| DN | affect, diagnos, defect, neuropathi, sex, diagnosi, drug, predict |
| ED | 50, dysfunct, primari, suscept, interv, new, sex |
| TyG+BMI | affect, triglycerid, adjust, women, bmi, autoimmun, particip, men |
| TyG+HGB | suscept, drug, longterm, autoimmun, hemoglobin, loss, triglycerid, primari |
| TyG | play, particip, class, triglycerid, loss, outcom |
| DN | nerv, neuropathi, dysfunct, content, liver, chronic, vitro, defect |
| UTI | vivo, excret, dysfunct, analys, urinari, play, support |
| Clinical Practice | 50, support, particip, hypothesi, intervent, drug, singl, hemoglobin |
| DR | affect, sex, retinopathi, loss, vivo, vitro, diagnos, defect |
| Clinical Practice | primari, play, intervent, defect, particip, vitro, hemoglobin, drug |
| ED | diagnos, play, dysfunct, vivo, loss, drug, particip, shown |
| NAFLD | affect, triglycerid, hepat, suscept, outcom, autoimmun, need, fatti |
| HGB+FSD | hemoglobin, adjust, triglycerid, dysfunct, drug, protect, women, sex |

Table 13: Display of top eight words in fourteen learned latent topics from DGLFRM in Pubmed dataset. The topic labels are abbreviations of clinical subareas related to Diabetes Mellitus. They are highly related to the top eight stemmed keywords.

shows the classification performance. We can see that SVM trained with node representation learned by NOARD achieves higher accuracy than other baselines, thus demonstrating the effectiveness of our model on downstream node classification task.

| | VGAE | KernelGCN | DGLFRM | VGNAE | NORAD |
|---|---|---|---|---|---|
| Cora | 65.24±0.73 | 78.21 ±1.53 | 83.70 ±1.85 | 83.74±0.92 | **84.79±1.12** |
| Citeseer | 57.09±0.25 | 64.99 ±2.02 | 69.10±2.70 | 74.79±1.11 | **74.04±0.98** |
| Pubmed | 85.57±0.41 | 80.20 ±1.37 | 78.53±1.59 | 86.36±0.46 | **87.91±0.29** |
| DBLP | 77.85±1.19 | 80.17±0.66 | 76.83±2.47 | 84.10±0.39 | **84.87±0.46** |

Table 14: Performance comparison of all models in the downstream classification task: We used an unpaired t-test to compare models' performance values. Not significantly worse than the best at the 5% significance level are bold.

