# OpenReview forum: "Interpretable Node Representation with Attribute Decoding"
_TMLR — Accepted by TMLR_

### Review · Reviewer_R5jC · 2022-07-13

**Summary Of Contributions:**

This paper proposes a set of modifications to the standard variational graph autoencoder model by Kipf and Welling, aiming to increase its performance and interpretability.

The main architectural contributions are the following:

1. Add a dedicated feature decoder
2. Change the latent prior to a sparse distribution (Bernoulli-like)
3. Add a procedure to separately deal with disconnected nodes, aiming to improve their attribute and connectivity recovery
4. Change the architecture and losses of the encoder and decoders to be more powerful

The authors show that the proposed architecture provides some advantages in two tasks and four datasets. The experiments also include an ablation study of the proposed architectural improvements to show that they are all beneficial.
Finally, the authors study the interpretability of the learned node attributes to recover the semantic meaning of the communities identified by the model. They show that the computed attributes have a nice semantic clustering.

**Requested Changes:**

## Typos
- Page 3: "let ... denotes the mutual information" -> denote
- Page 3: "this makes the encoder has no incentive" -> this means (?)

## Miscellaneous issues

- The paragraph about fixing the bias in link prediction is unclear to me. The training adjacency matrix will surely contain zeros that represent missing edges and non-edges, but this distinction is known a priori and already in the VGAE paper they used negative sampling to only sample the correct non-edges as negative samples.
I don't understand the reasoning behind what the authors claim on page 3.

**Strengths And Weaknesses:**

# Strengths
The paper proposes reasonable modifications to a well-known architecture, studying existing solutions in the literature and comparing them under a common theoretical framework.
All contributions are motivated and tested appropriately through ablation studies.
The paper is well-written and easy to follow, all concepts needed to understand the work are explained where appropriate.

Overall, I think the paper is in good shape besides some improvements that could be made to the experiments section (See below).


# Weaknesses

I only have comments related to the experiments:

- It is not clear if the baselines have a number of parameters comparable to the proposed method. Does the extra performance come at a significant computational cost? If so, this should be quantified and discussed.

- The experiments are conducted on very few and small datasets. The GNN community has long moved past the citation networks, so I advise the authors run their experiments on better datasets like OGB or similar. As it is, the results are unlikely to transfer to practical/non-benchmark problems.

- The experiment on interpretability should also compare the interpretability of the other baselines. As it is, it is not clear if the proposed method is actually more interpretable than the baselines.

---

> ### Author Response · Authors · 2022-08-17
> **Thank you for your comments!**
>
> We sincerely appreciate your feedback on our work. Please check our answers to your questions below.
>
> **Q1.** _It is not clear if the baselines have a number of parameters comparable to the proposed method. Does the extra performance come at a significant computational cost?_
>
> **A1.** For the encoder, since NORAD uses a shallower but wider GCN, it tends to contain slightly more parameters compared to the baselines. However, we argue that most of the parameters can be pruned out as the learned representations — community vector $C$ is sparse, which indicates that most of the model parameters are not effective. Also we find that increasing the parameters of the baselines does not improve the model performance. We have included the detailed discussion about the network parameters in Appendix A.2.
>
> On the other hand, the extra parameters of the decoder do not contribute to the computation. Because the decoders come with the purpose of providing better guidance of the encoding quality, and the examination of the quality of node representations only relies on the encoder.
>
> **Q2.** _I advise the authors to run their experiments on better datasets like OGB or similar._
>
> **A2.** We examine the performance of NORAD on three additional recent large-scale datasets, which can be found in the general comments.
>
> **Q3.** _The experiment on interpretability should also compare the interpretability of the other baselines. As it is, it is not clear if the proposed method is actually more interpretable than the baselines._
>
> **A3.** In general, the interpretability of our model can be analyzed via the prior, the node decoder, and the block model. The prior adopted by all baselines except DGLFRM [1] do not enable the learned representations with good interpretability. So we analyze the interpretability of the baselines by inspecting the shape of the embedding distribution. We provide the t-SNE visualizations for each baseline (Figure 5 in Appendix) to illustrate that NORAD learns more separable embeddings compared with the baselines. We also add the topic analysis of DGLFRM on Pubmed dataset, which can be found in the general comment.
>
> The preprocessing procedure is the same as we adopted in our model topic analysis (Section 5.4). There are 14 topics learned from the Pubmed dataset, and 3 of the topics have less than 8 words after removing the common high frequency words. We find that the learned topics by DGLFRM are not well separated and represented given the high frequency words: there exists topic repeating (e.g. Erectile Dysfunction (ED) appears twice), and topic overlapping (e.g. Hemoglobin (HGB) + Triglyceride (TyG) appears together).
>
> We analyze the reason why NORAD is more interpretable: the node representations learned by NORAD are used directly for link prediction via inner product Eq (10). However in DGLFRM, the node representations are first fed into a non-linear transformation (MLP), then the inner-product is applied for link prediction, the introduced nonlinearity dampens the interpreterbility of the learned node representations.
>
> **Q4.** _Typos._
>
> **A4.** We thank the reviewer and have corrected them.
>
> **Q5.** _The paragraph about fixing the bias in link prediction is unclear to me. The training adjacency matrix will surely contain zeros that represent missing edges and non-edges, but this distinction is known a priori and already in the VGAE paper they used negative sampling to only sample the correct non-edges as negative samples. I don't understand the reasoning behind what the authors claim on page 3._
>
> **A5.** We thank the reviewer for asking. First we'd like to politely point out that VGAE does not sample only the true negatives (correct non-edges). The edge masking code of VGAE is in https://github.com/tkipf/gae/blob/master/gae/preprocessing.py#L32. All the positive test edges are treated as false negative edges in the training process, and regarded as negatives during optimization.
>
> Those false negatives cause bias in model fitting. In our manuscript, we claim that when "$X$ and $A$ have a strong correlation, and $X$ is observed for all graph nodes. In this case, encoding information in $X$ helps improve the link prediction performance".
>
> ## Reference
> [1] Mehta, Nikhil, Lawrence Carin Duke, and Piyush Rai. "Stochastic blockmodels meet graph neural networks." International Conference on Machine Learning. PMLR, 2019.

---

> > ### Comment · Reviewer_R5jC · 2022-09-13
> > **Reply**
> >
> > I thank the authors for adding the experiments as requested.
> > I already had an overall positive opinion of the paper and the new results consolidate it.
> >
> > I can see this paper being of interest to the graph machine learning community and there are no concerns with the technical soundness of the work. I will change my recommendation to acceptance.

---

### Review · Reviewer_xbLj · 2022-07-22

**Summary Of Contributions:**

This paper makes a systematic analysis of VGAEs and shows that attribute decoding is important for node representation learning. The authors further propose a new learning model, NORAD, which encodes node representations in an interpretable approach.

**Broader Impact Concerns:**

No concerns about the ethical implications of the work.

**Requested Changes:**

1 Explore more challenging datasets:
- The link prediction tasks are too easy to solve. All the baseline models almost got perfect performance. The authors should explore more challenging datasets, for example, OGB link prediction tasks.
2 Add experiments for more types of tasks:
- In addition to unsupervised learning tasks, the authors should demonstrate that the learned node embeddings are helpful. A good experiment to explore is to use the fixed node embeddings from NORAD, and train simple classifiers for standard node classification tasks.

**Strengths And Weaknesses:**

Strengths:
- The model design for NORAD is technically sound.
- The contribution of having an interpretable unsupervised node representation learning method is valuable.

Weakness:
- The datasets used in the experiments are too limited.
- Demonstration of the usefulness of the model is missing.

---

> ### Author Response · Authors · 2022-08-17
> **Thank you for your comments!**
>
> We sincerely appreciate your feedback on our work. Please check our answers to your questions below.
>
>
> **Q1.** _The authors should explore more challenging datasets, for example, OGB link prediction tasks._
>
> **A1.** We examine the performance of our model NORAD on three additional recent large-scale datasets: OGBL-collab, OGBN-arxiv, Wiki-cs. The performance on the link prediction task is shown in the Tables: OGBL-collab, OGBN-arxiv, Wiki-cs. The results analysis can also be found in the general comments.
>
>
>
> **Q2.** _Add experiments for more types of tasks -- In addition to unsupervised learning tasks, the authors should demonstrate that the learned node embeddings are helpful. A good experiment to explore is to use the fixed node embeddings from NORAD, and train simple classifiers for standard node classification tasks._
>
> **A2.** To illustrate the usefulness, we use the node embeddings learned by NORAD for both the node classification and node clustering tasks. The performance of node classification is examined by ACC, which can be found in the Table Downstream task. The performance of node clustering is examined by NMI and ACC as we also used in the paper Table 2. We also show the node clustering performance on the two additional datasets in Table OGBN-arxiv and Table Wiki-cs.
>
> We can observe that the node representations learned by NORAD outperform the other baselines in the downstream tasks. This shows that the learned node representations are useful in downstream tasks, including supervised (classification) and unsupervised (clustering). At the same time, the node representations learned by NORAD exhibit better interpretability compared to other baselines.

---

### Review · Reviewer_CC4h · 2022-07-27

**Summary Of Contributions:**

This work proposes a Probabilistic Matrix Factorization method that uses an inductive (equivariant) GNN as a proposal distribution for the node embedding averages and variances. The method is based on Bayesian hierarchical models with spike and slab priors, so that we node embeddings are separated from the nodes' community assignments.

**Broader Impact Concerns:**

No concerns.

**Requested Changes:**

Please address Weaknesses 1,2,3.

**Strengths And Weaknesses:**

Strengths:
1. The paper is well-written. The introduction to probabilistic factor models of graphs is informative.
2. It is nice to see someone use the vast literature on Probabilistic Matrix Factorization to propose a graph model. There is a lot there that has not been properly explored in the context of graph representation learning.

Weaknesses:
1. The implication that the spike and slab distribution creates an "interpretable" distribution is not grounded in a clear notion of interpretation and how humans would interact with such model. The proposed node embedding Z_i is effectively a mixture model where a point (Z_i) can belong to multiple subpopulations (communities). For two nodes that belong to the same subpopulations (communities), the link is predicted based on the inner product between the embedding vectors, which is as interpretable as other matrix factorization methods. If i and j have overlapping communities and, say, we have K=100 communities (recall $v_{ik}$ is Gaussian). I am not sure how someone looking at the output of Eq (10) and the vectors $c_i, v_i$ and $c_j,v_j$ can categorically say "I understand exactly why the link (i,j) was predicted".  There are a few cases where I can see why the output would be interpretable. For instance, if i and j do not share any communities or just share one community. The argument that one can interpret the model would have been stronger had $v_{ik}$ being strictly positive.
2. One of the more puzzling parts of the model comes at Eq (13). The gnn(A,X;\phi) seems to be equivariant, such that two isomorphic nodes on the attributed graph (A,X) will get the same node embedding average and variance. This equivariant node representations is incompatible with link prediction in symmetric graphs (as theoretically described by https://arxiv.org/abs/1910.00452). The authors need to show why the concerns in https://arxiv.org/abs/1910.00452 about equivariance node representations for link prediction are not an issue in this model.
3. The authors need to better describe why it is OK to have an inductive model (GNN) output the mean and standard deviation of the embeddings while the community assignments and actual node embeddings (c_i) are both transductive. If the method is transductive, couldn't we also perform a transductive learning procedure with a hierarchical model (so that c_{ik} for all nodes in community k have the same prior Normal(mu_k,sigma_k))? It is unclear why we need to use the GNN.

---

> ### Author Response · Authors · 2022-08-18
> **Thank you for your comments!**
>
> We sincerely appreciate your feedback on our work. Please check our answers to your questions below.
>
> **Q1.** _interpretability of predicted links._
>
> **A1.** We would like to add more details about our implementation and the learned model.
>
> First, in our implementation we require $\mathbf{B}$ to be *symmetric* and *its diagonal elements to be 1*. We only learn the lower-triangle of $\mathbf{B}$.
>
> Second, in our result, the scale of off-diagonal elements of $\mathbf{B}$ are mostly below 0.1. Therefore, the connection of two nodes is still mainly decided by whether they have the same memberships in different topics.
>
> Third, we'd like to point out that the embedding vector is not a probability vector. The difference is similar to the difference between logits (taking positive and negative values) and categorical probabilities (only positive values). We use both positive and negative values in embedding vectors to get negative logits before the sigmoid function; otherwise $\mathbf{z}_i^\top \mathbf{B} \mathbf{z}_j$ is likely to be over zero, and the connection probability is over 0.5. In our case, a negative entry in the embedding vector also means that the node does not belong to the corresponding topic but has a stronger sense than a zero entry.
>
> Overall, we think our embedding vectors have a similar level of interpretability as sparse probability vectors.
>
> **Q2.** _equivariant node representations are incompatible with link prediction in symmetric graphs._
>
> **A2.** We agree with this point. However, our work mainly considers large networks that have rich node features (e.g. each node has a bag of words). In these cases, it is unlikely to get symmetric graphs. Furthermore, we are considering a general method, which means more powerful GNNs (e.g. GNNs using positional encoding) can be used to overcome this weakness.
>
> **Q3.** _why it is OK to have an inductive model (GNN) output the mean and standard deviation of the embeddings._
>
> **A3.** We use an inductive model for two reasons: amortization and link prediction. With a GNN, the learned encoding of one node can be used at a different node. This amortization idea greatly reduces the number of trainable parameters and speeds up training. In the task of link prediction, we do require the model to have the ability of generalization. The model needs to be able to discover new links by using the knowledge of existing links. A transductive model may "overfit" the networks and cannot accurately predict new links.

---

### Author Response · Authors · 2022-08-17
**Added Experiments (1/2)**

## 1.Experiments on additional benchmark datasets.

We follow the requests from the reviewers and examine our model on three more benchmark datasets, including OGBL-collab, OGBN-arxiv, and Wiki-cs [1,2]. For OGBL-collab, we perform the link prediction task and report the metric Hits@K, K=10, 50, 100 following [3]. For OGBN-collab, we perform link prediction and node clustering tasks following the experiment setting in our manuscript. For Wiki-cs, since all models yield high performance by following the data splitting strategy in VGAE as mentioned in [2], we lower down the train ratio and use split ratio: train/val/test: 0.6/0.1/0.3, we perform link prediction and node clustering tasks.

We report the mean and standard deviation of all evaluation metrics over 10 random data splits. The experiments results are shown as below.

We observe that NORAD outperforms most of the baselines on the three datasets. This shows that the model can also perform well on large-scale datasets besides the citation networks. (kernelGCN [4] requires transformation on the adjacency matrix, the computation memory cost is too high to scale it to large networks e.g. OGB datasets.)

### OGBL-collab

|         | VGAE | KernelGCN | DGLFRM | VGNAE | NORAD |
|:-:|-|-|-|-|-|
| Hits@10&uarr;  |  26.98 +- 0.12  |  NA  |  28.75 +- 0.68  |  31.24 +- 0.19  |  31.56 +- 0.23  |
| Hits@50&uarr;  |  44.76 +- 0.98   |  NA  |  46.19 +- 1.37  |  46.76 +- 0.88  |  47.38 +- 1.09  |
| Hits@100&uarr; |  51.44 +- 0.7  |  NA  |  51.49 +- 0.15  |  51.39 +- 0.42  |  51.40 +- 0.25  |

### OGBN-arxiv

|         | VGAE | KernelGCN | DGLFRM | VGNAE | NORAD |
|:-:|-|-|-|-|-|
| AUC&uarr; |  87.2 +- 1.37  |  NA  |  91.2 +- 0.38  |  92.3 +- 5.03  | 94.5 +- 0.22  |
| AP&uarr; |  88.7 +- 1.17  |  NA  |  92.0 +- 0.31  |  92.8 +- 4.12  |  94.6 +- 0.20  |
| NMI&uarr;|  7.26+-2.15  |  NA  |  9.35 +- 2.55  |  9.01 +- 2.15  |  10.32+-1.64  |
| ACC&uarr; |  10.25+-1.15  |  NA  |  11.19 +- 0.89  |  10.38 +- 0.97  |  11.13+-0.69  |

### Wiki-cs

|         | VGAE | KernelGCN | DGLFRM | VGNAE | NORAD |
|:-------:|------|-----------|--------|-------|-------|
| AUC&uarr; |  92.1 +- 0.99  |  93.1 +- 0.85  |  92.8 +- 0.90  |  95.3 +- 0.43  |  96.0 +- 0.22  |
| AP&uarr; |  91.5 +- 0.94  |  93.0 +- 1.22  |  93.0 +- 1.07  |  94.5 +- 0.55  |  96.0 +- 0.23  |
| NMI&uarr; |  21.2 +- 5.66  |  34.1 +- 5.20  |  20.4 +- 6.69  |  24.6 +- 1.10  |  34.1 +- 1.57  |
| ACC&uarr; |  29.9 +- 3.63  |  39.1 +- 3.86  |  30.4 +- 4.33  |  33.7 +- 1.01  |  39.4 +- 2.24  |

### Dataset summary
We show the summary of the three additional datasets, which are much larger than the datasets we used in the manuscript.
|         | Nodes | Edges  | Classes |
|:-------:|------|-----------|-------|
| OGBL-collab |  235,868  |  1,285,465  |  NA  |
| OGBN-arxiv |  169,343  |  1,166,243    |  40  |
| Wiki-cs |  11,701  |  216,123  |  10  |


## 2. Additional experiments on downstream tasks.
We follow the request from Reviewer xbLj and use the learned node representation to train classifiers for node classification tasks. The classifier we used here is SVM with RBF kernel. Node classification results are reported using ACC.

|         | VGAE | KernelGCN | DGLFRM | VGNAE | NORAD |
|:-------:|------|-----------|--------|-------|-------|
| Cora |  65.24 +- 0.73 |  78.21 +- 1.53  |  83.70 +- 1.85  |  83.74 +- 0.92  |  84.79 +- 1.12  |
| Citeseer |  57.09 +- 0.25  |  64.99 +- 2.02  |  69.10 +- 2.70  |  74.79 +- 1.11  |  74.04 +- 0.98 |
| Pubmed |  85.57 +- 0.41  |  80.20 +- 1.37  |  78.53 +- 1.59  |  86.36 +- 0.46  |  87.91 +- 0.29 |
| DBLP |  77.85 +- 1.19  |  80.17 +- 0.66  |  76.83 +- 2.47  |  84.10 +- 0.39  |  84.87 +- 0.46  |


## References
[1] Wang, Kuansan, et al. "Microsoft academic graph: When experts are not enough." Quantitative Science Studies 1.1 (2020): 396-413.

[2] Mernyei, Péter, and Cătălina Cangea. "Wiki-cs: A wikipedia-based benchmark for graph neural networks." arXiv preprint arXiv:2007.02901 (2020).

[3] Hu, Weihua, et al. "Open graph benchmark: Datasets for machine learning on graphs." Advances in neural information processing systems 33 (2020): 22118-22133.

[4] Tian, Yu, et al. "Rethinking kernel methods for node representation learning on graphs." Advances in neural information processing systems 32 (2019).

---

### Author Response · Authors · 2022-08-17
**Added Experiments (2/2)**

## 3. Topic analysis for baselines

We follow the request from Reviewer R5jC and add the topic analysis of the baselines. However, the baselines except DGLFRM don't enable interpretability so we only provide the topic analysis for DGLFRM. Here we again use Pubmed dataset. The preprocessing procedure is the same as we adopted in our model topic analysis (see Section 5.4).

| Topic label | Top Eight Stemmed Keywords |
|-------|------|
| Autoimmune Hepatitis (AIH) |  vivo, liver, releas, defect, hepat, given, content, autoimmun  |
| Diabetic Neuropathy (DN) |  affect, diagnos, defect, neuropathi, sex, diagnosi, drug, predict  |
| Erectile Dysfunction (ED) |  50, dysfunct, primari, suscept, interv, new, sex  |
| Triglyceride glucose-body mass index (TyG-BMI) |  affect, triglycerid, adjust, women, bmi, autoimmun, particip, men  |
| Hemoglobin (HGB) + Triglyceride (TyG) |  suscept, drug, longterm, autoimmun, hemoglobin, loss, triglycerid, primari  |
| Triglyceride (TyG) |  play, particip, class, triglycerid, loss, outcom  |
| Diabetic Neuropathy (DN) | nerv, neuropathi, dysfunct, content, liver, chronic, vitro, defect  |
| Urinary Tract Infection (UTI) | vivo, excret, dysfunct, analys, urinari, play, support  |
| Clinical practice | 50, support, particip, hypothesi, intervent, drug, singl, hemoglobin  |
| Diabetic Retinopathy (DR) | affect, sex, retinopathi, loss, vivo, vitro, diagnos, defect  |
| Clinical practice | primari, play, intervent, defect, particip, vitro, hemoglobin, drug  |
| Erectile Dysfunction (ED) | diagnos, play, dysfunct, vivo, loss, drug, particip, shown  |
| Nonalcoholic Fatty Liver Disease (NAFLD) | affect, triglycerid, hepat, suscept, outcom, autoimmun, need, fatti  |
| Hemoglobin (HGB) + Female sexual dysfunction (FSD) | hemoglobin, adjust, triglycerid, dysfunct, drug, protect, women, sex  |

As we have pointed out in the response to Reviewer R5jC's question 3, topics learned by NORAD are more interpretable than those by DGLFRM. The topics learned by NORAD are more separated and representative while those learned by DGLFRM are overlapped and repeated.

Our model is more interpretable than DGLFRM because we directly use the bilinear form to recover the adjacency matrix $A$ from node representations in Eq (10). DGLFRM transforms node representations using non-linear MLPs, and its representations are less interpretable. Please refer to A3. response to Reviewer R5jC for more details.

---

### Author Response · Authors · 2022-08-18
**Revised manuscript uploaded and responses to reviewers' comments posted**

Dear Reviewers,

We thank you for your valuable comments and suggestions on our manuscript. They are very constructive in improving our work. All the responses are posted. The experiments requested by the reviewers are updated in both the response and the revised manuscript. Please feel free to comment if there are any further questions or thoughts. We are looking forward to your further feedback.

Thanks for your efforts!

Best regards,

---

### Decision · Action_Editors · 2022-09-25

**Recommendation:** Accept with minor revision

**Comment:**

The paper proposes an alternative approach to generate node embeddings that are more interpretable than other approaches proposed in the literature. Not all reviewers agree that the generated embeddings are (easily) interpretable, and this is true in general terms. However, the paper shows evidence that the generated embeddings are more interpretable than the ones obtained by other approaches. I think this is an interesting contribution. This improvement is mainly obtained  thanks to the adoption of a modified loss function jointly with a graph modelling exploiting a probabilistic matrix factorization approach. Both choices impose a bias whose implications have not been fully discussed in the paper, and they constitute one of the major concerns by one reviewer. Considering the nature of TMLR, whose aim is to provide a fast but in-depth revision of short manuscripts, I think the current manuscript can be accepted provided that a careful revision of the paper explicitly discussing such implications, also in relation to other relevant papers (such as the appropriately mentioned paper https://arxiv.org/abs/1910.00452), is prepared. The development of a full theory formally unveiling the family of graphs that are properly modelled by the proposed approach can be the subject of an extended version.

---

> ### Author Response · Authors · 2022-11-10
> **Thank you so much for the recommendation and the feedback!**
>
> Thanks again for the recommendation and the feedback. We have made the following revisions according to the comment in the camera-ready version.
> 1. We have added two paragraphs explaining how our model structures (OSBM and ATN) help to learn interpretable structures.
> 2. We also cite the paper by Srinivasan & Ribeiro. The paper discussed the limitation of link prediction using structural node representations. Our work focuses on graphs with rich node attributes. Node attributes break node symmetries and help to overcome this limitation. Furthermore, node attributes also provide rich information about links in node representations. We have added a brief discussion of this point in the introduction section.
> 3. We merged all extra experiments required by the reviewers into the manuscript.
>
> Please check our revisions and let us know if there are more changes you want to see. Thank you!